# Will China's fertilizer use continue to decline? Evidence from LMDI analysis based on crops, regions and fertilizer types

Yuanmeng Ji, Huajun Liu⊙*, Yin Shi

School of Economics, Shandong University of Finance and Economics, Jinan, China

* huajun99382@163.com

**Data Availability Statement:** The national, provincial and all kinds of fertilizer use data are from China Rural Statistical Yearbook (https://data.cnki.net/yearbook/Single/N2019120190). The data names are 3-7 fertilizer application amount, 3-11

## Abstract

China implemented the Action Plan for the Zero Increase of Fertilizer Use in 2015, which led to a decrease in fertilizer use. However, Will fertilizer use continue to reduce? With data obtained from 2006 to 2017, the paper used the logarithmic mean Divisia index (LMDI) method to analyze the scale effect, intensity effect and structural effect of fertilizer use change in China from three aspects: crops, regions and fertilizer types. Our finding suggests that (1) The intensity effect was the most critical factor affecting the decline in fertilizer use in China. (2) The sowing scale and fertilization intensity of grain, vegetables and fruits had the most significant driving effect on fertilizer reduction. (3) The three effects of each region were different in space, and the eastern region contributed most to the fertilizer decrement. (4) Nitrogen fertilizer and compound fertilizer had the most considerable influence on fertilizer reduction, especially in the sowing scale and fertilization intensity since 2009. The government should establish a fertilizer reduction management system, which includes scale control, intensity reduction, structural adjustment and other measures.

## 1. Introduction

China's population reached 1.386 billion by 2017, accounting for 18.7% of the world's population [1]. For such a large community, the question of providing food is increasingly essential. However, China has 119,491.1 hectares of arable land, which is only 8.6% of the world's total [2]. Thus, it is an excellent feat for China to feed so many people [3, 4]. With an annual growth rate of 3.6%, China's food production increased from 43,069.5 million tons in 2003 to 66,060.3 million tons in 2015 and has successfully achieved 12 years of continuous growth [5]. The remarkable accomplishment of China's food production is mainly attributed to the abundant input of fertilizer [6, 7]. According to the data of the China Statistical Bureau, fertilizer use (FU) in China increased from 44.116 million tons in 2003 to 60.226 million tons in 2015, which accounted for more than one-third of the world's total amount (Fig 1). Nevertheless, the overuse of fertilizer has caused a series of harmful problems, such as low nutrient utilization rate and even soil loss [8], environmental pollution, and ecological damage [7, 9, 10].

In recent years, the Chinese government has recognized the seriousness of the overuse of fertilizer. It has put forward the decision of reducing the amount of fertilizers and increasing

agricultural fertilizer application amount (calculated by pure method) and 3-9 agricultural fertilizer application amount. The fertilization intensity at the crop level is derived from the National Agricultural Product Cost-Benefit Compendium (https://data.cnki.net/yearbook/Single/N2019120280), and named after "average fertilizer input." The sown area and yield per unit area of crops in each province are from the official website of the National Bureau of statistics of China (http://www.stats.gov.cn/) under the data set names "sown area of main crops" and "yield per unit area of main crops." Due to the lack of data, this paper considers that the fertilizing area of each fertilizer is equal and replaced by the planting area of crops, which also comes from the official website of China Statistics Bureau. The authors of the present study had no special access privileges in accessing these data sets which other interested researchers would not have.

**Funding:** This research was supported by The National Social Science Fund of China (Grant no. 18BJY140), The Natural Science Foundation of Shandong (Grant no.ZR2019MG007) and The Natural Science Foundation of Shandong (Grant no. ZR2019MG029). HL would like to acknowledge Taishan Scholar Project for youth experts (Grant no.tsqn20171208) and Special Support Plan for High-level Talents of Shandong University of Finance and Economics.The funders had no role in study design, data collection and analysis, decision to publish, or preparation of the manuscript.

**Competing interests:** The authors have declared that no competing interests exist.

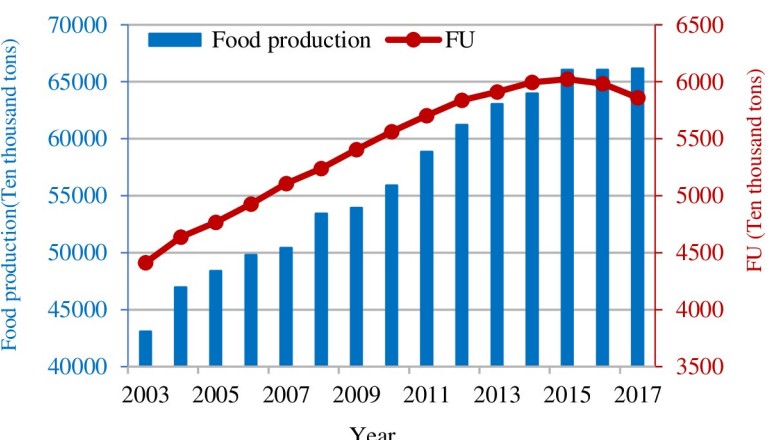

**Fig 1. Food production and FU in China during 2003–2017.**

the efficiency on the premise of stable food production growth and adequate protection of food security. In 2015, the Chinese government promulgated the Action Plan for the Zero Increase of FU, which proposed a goal of "zero growth of FU, the establishment of a scientific fertilizer management technology system, and the improvement of the scientific FU level". Then, in 2016 and 2017, Central Document No.1 noted that the "zero growth" action of fertilizer should be carried out. China's zero-growth action in FU has achieved initial results. In 2016, China's FU approached zero growth for the first time. FU in China declined from 60.226 million tons in 2015 to 58.59 million tons in 2017, with an annual rate of decline of 1.8% (Fig 1). Then, will the decline in China's FU be sustainable? Further research on this question not only helps us better explore the driving effects of China's fertilizer reduction and influence of each effect but also provides a more comprehensive reference for policymakers to continuously control the fertilizer decrement and develop a sound fertilizer reduction management system.

Existing literature on FU by global scholars mainly focuses on four aspects: (1) benefit evaluations of FU on crop yield [11–15]; (2) effects of FU on soil fertility and nutrients [12, 16–18]; (3) assessment of the damage caused by overfertilization on environmental ecology [8, 19–22]; and (4) research on scientific management strategies of FU [23–26]. These documents have laid a foundation for us to understand the reduction of fertilizer, but few studies have explored the source of fertilizer reduction. The principal reason is that FU did not begin to decline until 2015 when there was a shortage of samples for scholars to study. A few scholars have made efforts to explore the agricultural factors affecting the reduction of fertilizer. For example, Yang and Lin [27], based on panel data in 2002 and 2016, used the logarithmic mean Divisia index (LMDI) to study the driving factors and contribution rate of fertilizer reduction in Zhejiang Province; Cai et al. [28], based on statistical data from 2004 to 2015, divided 10 provinces of China's primary grain production into three regions by using the Laspeyres method and analyzed the influencing factors of FU intensity from the regional perspective. However, the above studies were still limited to a regional perspective. Furthermore, there has been some research on the sustainability of the decline in energy use, especially the decline in coal use [29, 30]. However, this topic has not been addressed for the field of fertilizers.

To comprehend diachronic changes in fertilizer decrement, assessing the prime factors that underlie the development of FU is essential [31]. According to previous research, structural decomposition analysis (SDA) and index decomposition analysis (IDA) are the two most

commonly used methods of factor decomposition [32–34]. SDA is based on input-output data in quantitative economics, while IDA uses aggregate data at the sector level [35, 36]. Boyd et al. [37] noted that IDA could clearly show the change in indicators over time. Thus, IDA is more suitable for this paper. LMDI has several advantages: no residuals in the analysis process, meets the molecular reversal test, fertilizer can be easily broken down into several items, available data are extensive, and zero value problems can be processed centrally [38–41]. The LMDI method, therefore, was applied in this study.

Based on the existing research, the main contributions of this paper are two aspects. First, this paper expands the perspective of fertilizer research and discusses the sources of fertilizer reduction from the perspectives of crop, region and fertilizer type. Second, this paper is the first attempt to answer the question of the sustainability of fertilizer reduction in China. Therefore, the paper used the LMDI method to decompose the driving factors of the change of FU in 2006–2017 from three aspects of crop, region and fertilizer type, and deeply explores the sources of fertilizer reduction in China from different perspectives. The purpose of the study is not only to provide scientific reference for better reducing usage, increasing efficiency of fertilizer and controlling the excessive use of fertilizer, but also to provide targeted policy suggestions for exploring modern fertilizer management and achieving the goal of "zero growth of FU" in China.

The remainder of the paper is organized as follows. Section 2 describes the LMDI decomposition method, data sources and processing from the perspectives of crops, regions and fertilizer types. Section 3 analyzes the driving factors of fertilizer reduction and discusses a more profound implication of the results from different perspectives. Section 4 concludes this article and presents policy suggestions for the sustainable development of fertilizer reduction.

## 2. Materials and methods

### 2.1 Decomposition method

In this study, the changes in FU in China were decomposed using the LMDI method from the perspectives of crops, regions and fertilizer types. Three-factor decomposition was proposed to quantify the main determinants of FU changes and analyze the effects of various influencing factors. The effects are the intensity effect (IE), structure effect (STE) and scale effect (SE), respectively.

**2.1.1 Crop decomposition.**   First, from the perspective of crops, China's total FU can be decomposed as follows:

$$F = \sum_{i=1}^{8} f_i = \sum_{i=1}^{8} \frac{f_i}{S_i} \times \frac{S_i}{S} \times S = \sum_{i=1}^{8} I_i \cdot ST_i \cdot S \tag{1}$$

where F is the total FU of 8 crops, and i represents the crop (tobacco, sugar, beans, cotton, oils, fruits, vegetables and grain). $f_i$ is the FU of crop i, $S_i$ is the sown area of crop i, and S is the total sown area of eight crops. $f_i/S_i$, $S_i/S$ and S are represented by $I_i$, $ST_i$ and S, respectively. $I_i$, $ST_i$ and S designate the intensity factor, structure factor and scale factor, respectively.

In this study, $F^T$ and $F^0$ are assumed to be the FU in the base year and t year, respectively. ΔF designates the variation from the base year to year t. According to the LMDI addition model, the equation can be expressed as follows:

$$\triangle F = F^T - F^0 = \triangle F_{I\_effect} + \triangle F_{ST\_effect} + \triangle F_{S\_effect} \tag{2}$$

where $\Delta F_{I\_effect}$, $\Delta F_{ST\_effect}$ and $\Delta F_{S\_effect}$ are the IE, STE and SE caused by the intensity factor,

structure factor and scale factor, respectively. The effects can be expressed as follows:

$$\triangle F_{I\_effect} = \sum_{i=1}^{8} \frac{F_i^T - F_i^0}{\ln F_i^T - \ln F_i^0} \ln\left(\frac{I_i^T}{I_i^0}\right) \tag{3}$$

$$\triangle F_{ST\_effect} = \sum_{i=1}^{8} \frac{F_i^T - F_i^0}{\ln F_i^T - \ln F_i^0} \ln\left(\frac{ST_i^T}{ST_i^0}\right) \tag{4}$$

$$\triangle F_{S\_effect} = \sum_{i=1}^{8} \frac{F_i^T - F_i^0}{\ln F_i^T - \ln F_i^0} \ln\left(\frac{S^T}{S^0}\right) \tag{5}$$

**2.1.2 Region decomposition.** Second, from the perspective of regions, China's total FU can be decomposed as follows:

$$F = \sum_{j=1}^{31} f_j = \sum_{j=1}^{31} \frac{f_j}{S_j} \times \frac{S_j}{S} \times S = \sum_{j=1}^{31} I_j \cdot ST_j \cdot S \tag{6}$$

where F is the total FU in 31 provinces, and j represents the province. $f_i$ is the FU of province j, $S_j$ is the sown area of province j, and S is the total sown area of 31 provinces. $f_j/S_j$, $S_j/S$ and S are represented by $I_j$, $ST_j$ and S. $I_j$, $ST_j$ and S designate intensity factor, structure factor and scale factor, respectively.

According to the LMDI addition model, according to the LMDI addition model, ΔF, IE, STE and SE equations can be further expressed as follows:

$$\triangle F = F^T - F^0 = \triangle F_{I\_effect} + \triangle F_{ST\_effect} + \triangle F_{S\_effect} \tag{7}$$

$$\triangle F_{I\_effect} = \sum_{j=1}^{31} \frac{F_j^T - F_j^0}{\ln F_j^T - \ln F_j^0} \ln\left(\frac{I_j^T}{I_j^0}\right) \tag{8}$$

$$\triangle F_{ST\_effect} = \sum_{j=1}^{31} \frac{F_j^T - F_j^0}{\ln F_j^T - \ln F_j^0} \ln\left(\frac{ST_j^T}{ST_j^0}\right) \tag{9}$$

$$\triangle F_{S\_effect} = \sum_{j=1}^{31} \frac{F_j^T - F_j^0}{\ln F_j^T - \ln F_j^0} \ln\left(\frac{S^T}{S^0}\right) \tag{10}$$

Also, we discussed the LMDI decomposition in four regions: eastern, northeastern, central and western. Eastern: Beijing, Tianjin, Hebei, Shandong, Jiangsu, Shanghai, Zhejiang, Fujian, Guangdong and Hainan; Northeast: Heilongjiang, Jilin and Liaoning; Central China: Shanxi, Henan, Hubei, Hunan, Jiangxi and Anhui; Western: Chongqing, Sichuan, Guangxi, Guizhou, Yunnan, Shaanxi, Gansu, Inner Mongolia, Ningxia, Xinjiang, Qinghai and Tibet. The formula is similar to the decomposition of 31 provinces (Omit specific decomposition steps).

**2.1.3 Fertilizer type decomposition.** Third, from the perspective of fertilizer types, China's total FU can be decomposed as follows:

$$F = \sum_{k=1}^{4} f_k = \sum_{k=1}^{4} \frac{F}{S} \times \frac{f_k}{F} \times S = \sum_{k=1}^{4} I \cdot ST_k \cdot S \tag{11}$$

where, F is the FU of 4 fertilizers, and k represents the fertilizer types (nitrogen fertilizer, phosphate fertilizer, potash fertilizer and compound fertilizer). $f_k$ is the FU of fertilizer k and S is the total sown area of 4 fertilizers. $F/S$, $f_k/F$ and S are represented by I, $ST_k$ and S designating intensity factor, structure factor and scale factor, respectively.

According to the LMDI addition model, ΔF, IE, STE and SE equations are as follows:

$$\triangle F = F^T - F^0 = \triangle F_{I\_effect} + \triangle F_{ST\_effect} + \triangle F_{S\_effect} \tag{12}$$

$$\triangle F_{I\_effect} = \sum_{k=1}^{4} \frac{F_k^T - F_k^0}{\ln F_k^T - \ln F_k^0} \ln\left(\frac{I^T}{I^0}\right) \tag{13}$$

$$\triangle F_{ST\_effect} = \sum_{k=1}^{4} \frac{F_k^T - F_k^0}{\ln F_k^T - \ln F_k^0} \ln\left(\frac{ST_k^T}{ST_k^0}\right) \tag{14}$$

$$\triangle F_{S\_effect} = \sum_{k=1}^{4} \frac{F_k^T - F_k^0}{\ln F_k^T - \ln F_k^0} \ln\left(\frac{S^T}{S^0}\right) \tag{15}$$

## 2.2 Data sources and processing

The research period for this paper was from 2006 to 2017, and this decision was based on the following considerations. First, since 2006, China's agricultural development has entered a new stage. The Chinese government implemented the "Eleventh Five-Year Plan" and "Twelfth Five-Year Plan" of national agricultural and rural economic development to standardize the fertilizer market. In 2015, the Ministry of Agriculture issued the Action Plan for the Zero Increase of FU to speed up the reform of the fertilizer market. Second, farmers' fertilizer applications have a definite "lack in" characteristic [42]. Considering the data availability, the analysis in the past 12 years can not only see the implementation degree of previous policies but also provide a reference for future policy formulation.

The national, provincial and all kinds of fertilizer use data are from China Rural Statistical Yearbook (https://data.cnki.net/yearbook/Single/N2019120190). The data names are 3–7 fertilizer application amount, 3–11 agricultural fertilizer application amount (calculated by pure method) and 3–9 agricultural fertilizer application amount. The fertilization intensity at the crop level is derived from the National Agricultural Product Cost-Benefit Compendium (https://data.cnki.net/yearbook/Single/N2019120280). And named after "average fertilizer input". The sown area and yield per unit area of crops in each province are from the official website of the National Bureau of statistics of China (http://www.stats.gov.cn/) which names are "sown area of main crops" and "yield per unit area of main crops". Due to the lack of data, this paper considers that the fertilizing area of each fertilizer is equal and replaced by the planting area of crops, which also comes from the official website of China Statistics Bureau. In addition, I have confirmed that the authors of the present study had no special access privileges in accessing these data sets which other interested researchers would not have.

From the crop perspective, the main crops were divided into eight categories: grain, beans, oil, sugar, cotton, tobacco, fruits and vegetables, according to the instructions of the Rural Statistics Yearbook of China. Eight categories of sowing area represent the fertilization area. The average value of representative crops was used instead of the fertilization intensity of each crop: grain primarily contained wheat, maize and rice (early, middle, late and japonica rice); the center of beans was soybeans; peanut and rapeseed were the main oils; sugar was replaced by sugarcane and beet; tobacco was principally flue-cured tobacco and sun-cured tobacco;

fruits focused on citrus. The kernel vegetables were tomatoes, cucumbers, eggplants, cabbage, pepper, Chinese cabbage and potatoes. The amounts of fertilizer used in different crops was missing; thus, we estimated the FU of different crops by multiplying the fertilization intensity by the sown area.

For the problem of zero values in data, Ang and Choi [43] showed that a minimal number can replace the zero value. When tends to zero, the result is obtained by convergence. Since then, studies such as Ang and Liu [40] and Ang [44] have also applied this strategy. According to the operation of Xu et al. [35] and Wen and Li [45], we do the following:

$$\frac{F^T - F^0}{\ln F^T - \ln F^0} = \begin{cases} (F^T - F^0)/(\ln F^T - \ln F^0), F^T \neq F^0 \\ F^T, F^T = F^0 \\ 0, F^T = F^0 = 0 \end{cases} \tag{16}$$

## 3. Results and discussion

According to Formulas (1)–(16), the change in FU in China from 2006 to 2017 was decomposed into three driving factors, which could be divided into four stages. The purpose was to identify what caused the change and what section drove it. Detailed results and discussion are as follows.

### 3.1 Driving factors based on crops affecting fertilizer reduction

**3.1.1 Total effect based on crops.** In Fig 2A, the four phases are described as follows:

Phase I (2006–2008): The SE and IE promoted the increase in FU; additionally, only the STE decreased, and the total FU increased at this stage.

Phase II (2009–2011): To maintain rapid economic development and realize the steady increase in China's agricultural production, increasing the intensity of fertilizer application was a necessary measure. Also, China adjusted the structure of crops and expanded the planting area of cash crops with high fertilizer consumption, while the SE did not change significantly. At this stage, three positive effects contributed to the rapid increase in FU.

Phase III (2012–2014): The most obvious change compared to the previous phase was the rapid decline in IE. Environmental pollution received increasing attention. According to China's 12th Five-Year Plan, China's fertilizer orientation changed from "promoting development" to "promoting regulation", and this change focused on improving the utilization rate of fertilizer [6]. The SE decreased slightly. The degree of increase of fertilizer application decreased.

Phase IV (2015–2017): The FU decreased profoundly during this period, and this decrease was accompanied by the development goal of zero growth of fertilizer and the implementation of a series of measures. The SE and IE, from positive to negative, played a role in FU reduction for the first time. The STE continued to promote the reduction of fertilizer, and the extent of reduction was increased. Thus, the fertilizer reduction in China was developing well.

To further analyze the decomposition results of China's FU variation, several principal crops were selected to analyze their roles in the change in the SE, IE and STE, as shown in Fig 2B–2D and Fig 3. The eight crops included grain, beans, oils, sugar, cotton, tobacco, fruits and vegetables.

**3.1.2 Scale effect based on crops.** From 2006 to 2017, the SE generally declined in Fig 2B. The SE decreased slowly from 2006 to 2014, which was mainly caused by the decline in the SE of grain, vegetables and fruits. The contributions of other crops to the SE were mild. Moreover, all crop SE values changed from positive to negative, and this change represented a switch from promoting FU to reducing the use of fertilizer during the period from 2015 to 2017, ultimately causing a sharp decline in the SE. The reason for this reversal was that the demand for

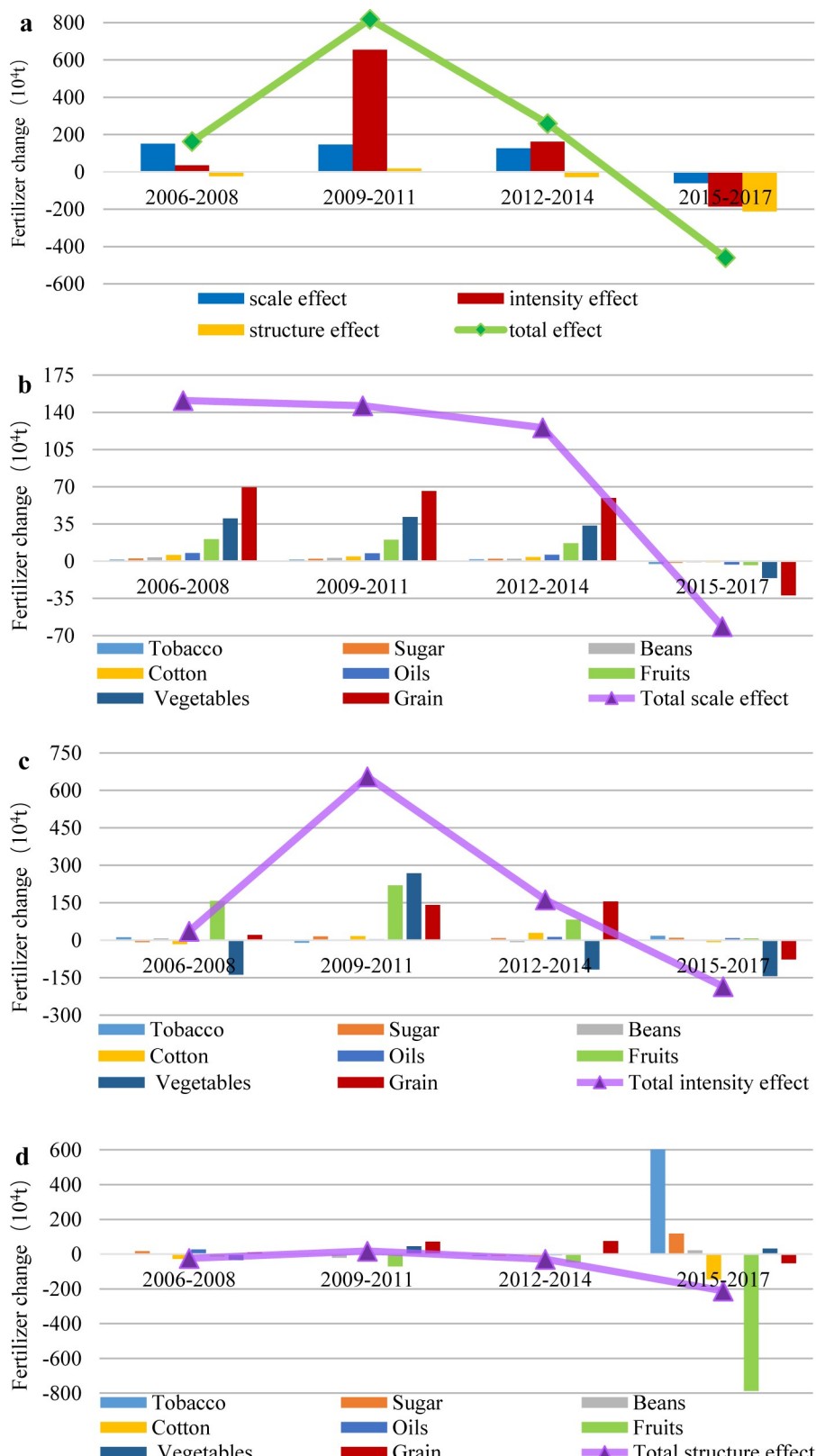

**Fig 2.** The total effect, scale effect, intensity effect and structure effect based on crop perspective: (a) Chinese FU variation decomposition based on crops;(b) Contributions of key crops to scale effect; (c) Contributions of key crops to intensity effect;(d) Contributions of key crops to structure effect.

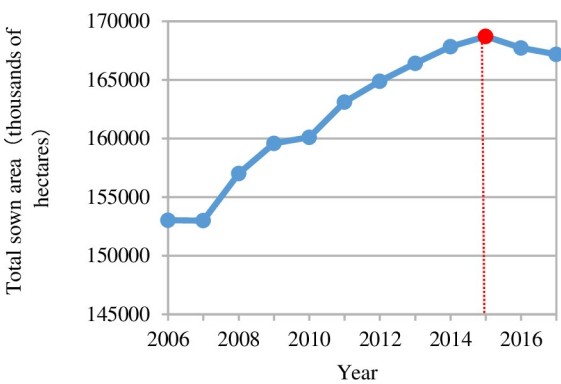

**Fig 3. Total sowing area based on crops during 2006–2017.**

scale was no longer sufficient to meet agricultural production [27, 6]. As shown in Fig 3, controlling the sowing scale and improving the FU efficiency is the correct solution to reduce the amount of fertilizer [46, 47].

**3.1.3 Intensity effect based on crops.** Fertilization intensity is measured by the amount of fertilizer applied per unit area. We believe that a higher IE indicates more fertilizer is applied. In Fig 2C, the FU increased by approximately 6 million tons in the period of 2006–2011, primarily due to the change in three crops (grain, vegetables and fruits). Vegetable intensity changed from a negative effect to a positive effect and became the dominant driving factor of fertilizer increment. The IE of fruits and grain also increased dramatically. The increase in FU began to decline or even displayed negative growth, and the decline in the IE of grain, vegetables and fruits played a vital role during 2012–2017. Theoretically speaking, on the premise that the quantity of agricultural products does not decline, the reduction in FU intensity indicates an improvement in energy efficiency, which is usually the result of technological progress [35]. Since the 12[th] Five-Year Plan, especially the 13[th] Five-Year Plan, the government has attached great importance to the work related to agriculture, vigorously promoted scientific and technological innovation, strengthened the necessary conditions for agricultural technology, and made great efforts to improve related science.

**3.1.4 Structure effect based on crops.** In Fig 2D, the variability of the STE and the contribution of all crops to the STE were imperceptible during 2006–2014. The real transformation took place between 2015 and 2017. Under the new situation, the contradiction in agriculture changed from an insufficient total amount to a structural discrepancy. It is a critical task for agricultural economies to push forward the supply-side structural reform of agriculture and accelerate the transformation of the agricultural development mode. In 2016, the Ministry of Agriculture issued the National Planting Industry Structure Adjustment Plan (2016–2020), making specific arrangements for the adjustment of agricultural structure. These arrangements included the following: ensured grain yield; stable cotton, oil and sugar; coordinate the production and demand of fruits and vegetables; beans were tailored to local conditions, but restrictions on tobacco were eased. In addition, the positive STE of tobacco was the largest, and its fertilizer increment was 6.031 million tons. The negative STE of fruits was the strongest, and its weight loss reached 7.881 million tons. The total STE promoted the fertilizer reduction.

**3.1.5 The yield of crops during fertilizer reduction.** In recent years, the three effects have driven the decline of FU in varying degrees. Especially after 2015, China's FU has successfully decreased year after year. On the contrary, China's food production has not been reduced, but continued to rise (Fig 1). So, from the perspective of crops, will the decrease of FU lead to the decrease of crop yield?

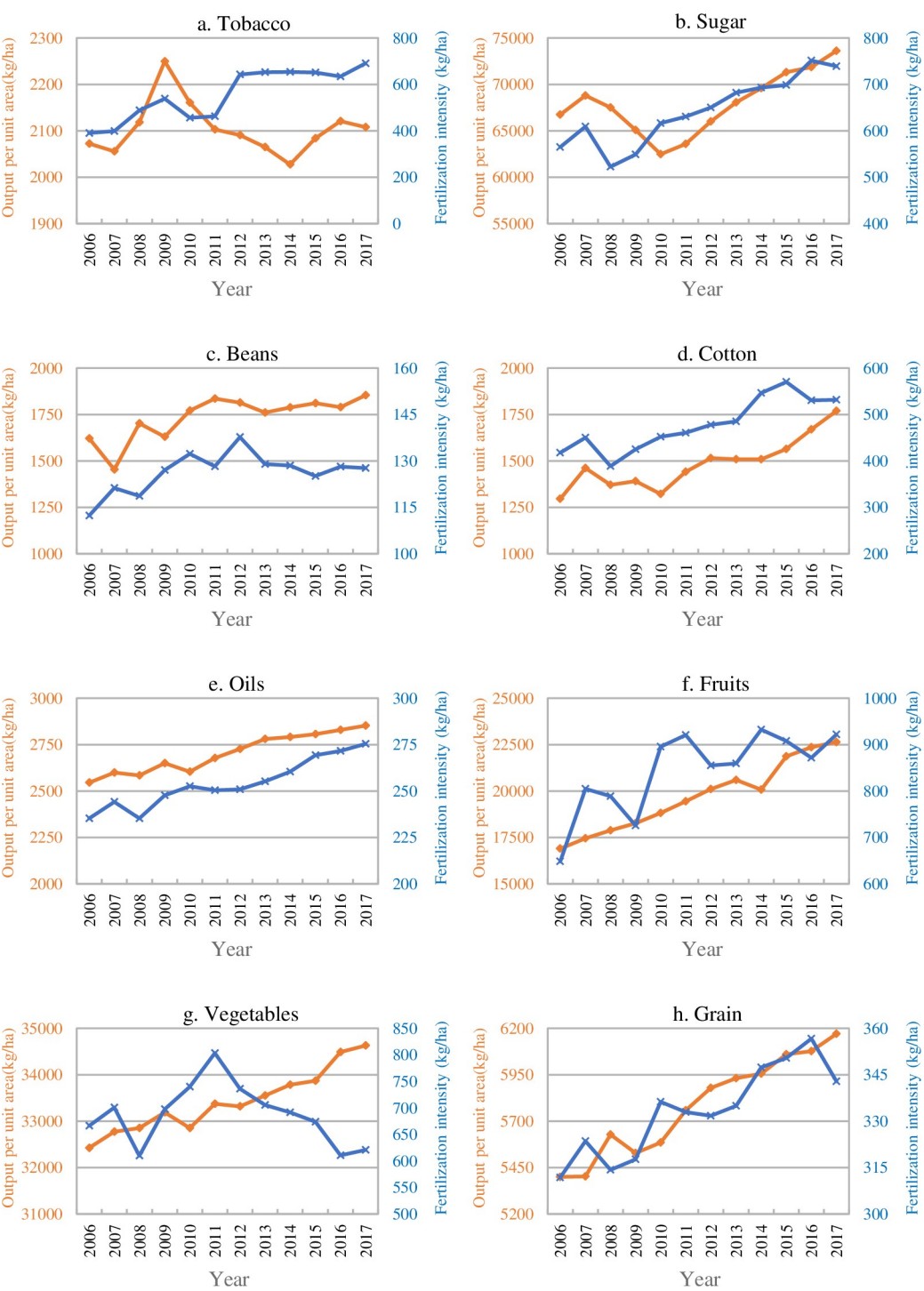

**Fig 4. Per unit yield and fertilization intensity of eight crops.**

Fig 2 shows that the IE is the most important factor leading to the change of crop FU, and also the main source of the decrease of FU, while vegetables, grain and fruits are the main crops causing the decrease of IE. It can be seen from Fig 4 that the fertilization intensity of vegetables, grain, sugar, beans and cotton has been reduced, especially grain and vegetables.

However, the unit yield of crops has not declined as a result, showing a continuous growth phenomenon. The main reasons for this are as follows. First, there is a general phenomenon of excessive fertilization in China's agriculture. Therefore, properly reducing the intensity of fertilization will not lead to the loss of nutrients in crops, which will not threaten crop yield. Second, according to different fertilization methods, China has developed some mature technical models, especially the promotion of high-yield and high-efficiency cultivation technology model, which can not only reduce the fertilizer intensity, but also increase the per unit yield to a certain extent. Third, the popularization of soil testing formula fertilization technology also plays a role in saving fertilizer and increasing production. In addition, although the fertilization intensity of fruits fluctuated greatly, it did not affect the growth of per unit yield. In the next step, we should continue to control the fertilization intensity of fruits. The fertilization intensity of oils is very similar to the change trend of per unit yield, and the application intensity of fertilizer is likely to have a high impact on per unit yield. Therefore, replacing conventional materials with new fertilizers may achieve the reduction of fertilizer without affecting per unit yield. It is worth noting that after 2009, the fertilization intensity and yield of tobacco changed in the opposite direction, indicating that the decline of fertilization intensity of tobacco will not directly lead to the decline of yield. In this view, China's FU reduction action is implemented under the condition of ensuring food security or crop production security. Fertilizer reduction will not lead to crop production reduction.

## 3.2 Driving factors based on regions affecting fertilizer reduction

There are spatial and temporal differences in FU in different regions. According to the value of the effect, this paper divided 31 provinces into four categories: strong negative effect, weak negative effect, weak positive effect and strong positive effect (a negative effect means that the effect leads to a decrease in FU, while a positive effect means that the effect leads to an increase in FU). The purpose was to determine which regions contributed more to the reduction in FU.

**3.2.1 Total effect based on regions.** Fig 5A illustrates FU change values, which reflect the variation in FU across four regions from 2006 to 2017. The STE and SE supported the increase in FU during 2006–2014. This result was due to the rapid development of urbanization in China [48]. China's urbanization rate grew at an average annual rate of 1.3 percentage points, and the urban population grew by an average of 17.587 million per year during 2006–2014. Urban land use continued to increase, while agricultural land development slowed. Therefore, increasing the intensity of fertilizer application became a universal way to ensure food security [49, 50]. However, the contribution of these two effects decreased, which indicated that the growth rate of fertilizer application decreased. The STE was weak in fertilizer decrement. In recent years, facing the increasing agricultural nonpoint source pollution in China, the intensity of the application has declined. Provinces have been strengthening agricultural infrastructure and improving the utilization efficiency of fertilizers. From 2015 to 2017, the IE and SE decreased significantly, and the IE even turned into a negative effect. The STE increased the degree of fertilizer reduction.

**3.2.2 Scale effect based on regions.** As shown in Fig 6 and Table 1, the SE was generally positive, and there were no negative driving provinces from 2006 to 2011. Most of the provinces had weak positive effects, and there were ten strong positive provinces, including Hebei, Jiangsu, Anhui, Shandong, Henan, Hubei, Hunan, Guangdong, Guangxi and Sichuan. Supported by a series of policies favoring agriculture [51], these traditional agricultural provinces actively expanded the scale of agricultural production and increased the amount of fertilizer. From 2012 to 2014, with the development of industrialization and urbanization, land utilization became more intensive [52], the eight provinces of Hebei, Jiangsu, Anhui, Hubei, Hunan,

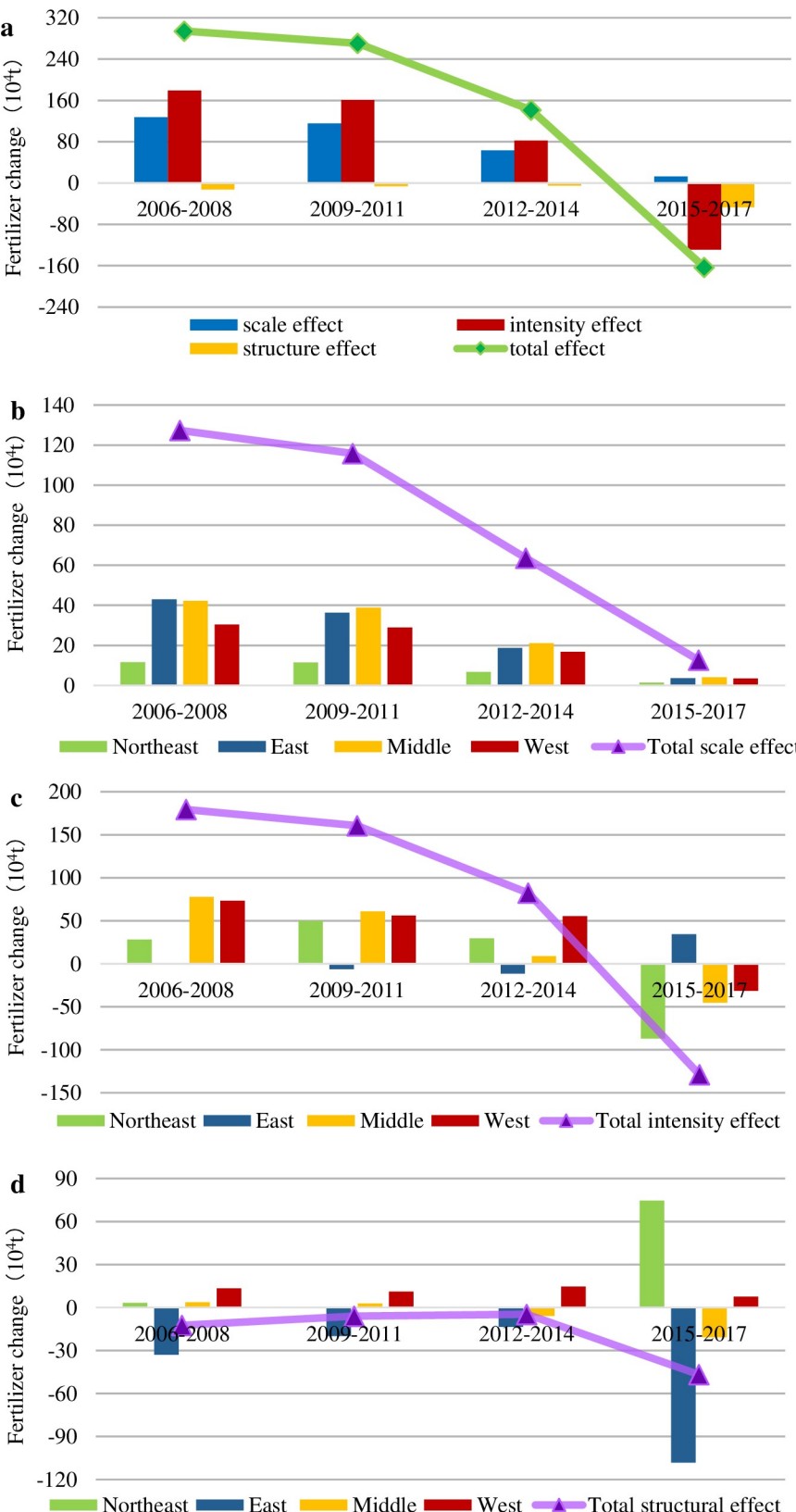

**Fig 5. The total effect, scale effect, intensity effect and structure effect based on regions perspective.** (a) Chinese FU variation decomposition based on regions;(b) Contributions of four regions to scale effect; (c) Contributions of

four regions to intensity effect; (d) Contributions of four regions to structure effect. because the data sources are different from the crop perspective, we re-decomposed the total FU effect.

Guangdong, Guangxi and Sichuan restricted the cultivated area, transforming from an active positive SE to a weak positive SE, while Shandong and Henan still showed a strong positive effect. After 2015, every province energetically adjusted and reduced the agricultural area, without exception, and this practice caused the value to transform to a negative SE and made a significant contribution to the reduction of fertilizer.

As seen in Fig 5B, the main reason for the decrease in the SE was the driving effect in the eastern and central regions, which indicated that the planting scale in the eastern and central regions had an evident trend and reduced the amount of fertilizer. In the west, the food pressure was relatively small due to the sparsely populated land. Thus, optimizing the cultivated land area can reduce the FU. In Northeast China, agriculture and heavy industry kept pace with each other, and the planting scale was relatively stable, showing a slight promoting effect on fertilizer application.

**3.2.3 Intensity effect based on regions.** Fig 6 and Table 1 show that this factor made the largest contribution to the decrease in FU in most provinces of China. Between 2006 and 2017,

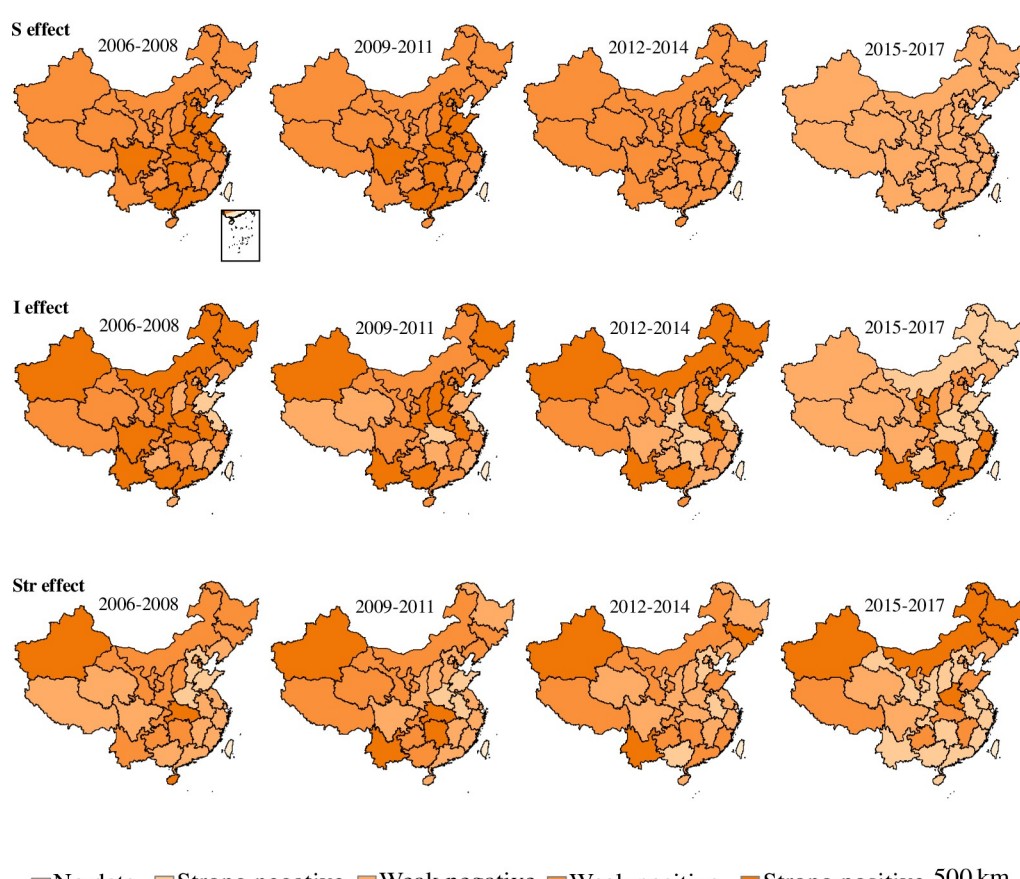

**Fig 6. Temporal and spatial differences in the driving effect of FU in China.** Figures are drawn by the authors according to the standard map of the National Surveying and Mapping Geographic Information Bureau (Approved drawing number: GS (2016) 2921) (http://bzdt.ch.mnr.gov.cn/). All maps on this website are available for free download without copyright.

**Table 1. Number of provinces with negative effects in each stage.**

| Effect | 2006–2008 | 2009–2011 | 2012–2014 | 2015–2017 |
|---|---|---|---|---|
| Scale effect | 0 | 0 | 0 | 31 |
| Intensity effect | 8 | 9 | 13 | 18 |
| Structure effect | 16 | 16 | 19 | 21 |

the number of provinces with a positive IE decreased, while the number of those with a negative IE increased. Within four periods, the figure of provinces with a negative IE was 8, 9, 13 and 18, respectively, which indicated that an increasing number of provinces realized that excessive FU intensity not only achieves yield increase but also hinders sustainable agricultural development.

As shown in Fig 5C, eastern regions promoted the reduction of FU during 2006–2014, especially in Shandong (-29.5×10$^4$ t) and Jiangsu (-26.9×10$^4$ t). These provinces had a high level of economic development, relatively advanced agricultural technology, and a high utilization efficiency of fertilizer, thus reducing fertilizer application [53, 47]. Unfortunately, the IE in the eastern regions increased from 2015 to 2017. In contrast, the positive IE was strong in the central and western regions, but the decline was noticeable, such as that in Inner Mongolia (-35.0×10$^4$ t), Hubei (-31.2×10$^4$ t) and Guizhou (-13.4×10$^4$ t). This result was because the early fertilizer intensity in the central and western provinces was prodigious, and the fertilizer intensity had ample space to decline. The IE fluctuated greatly in Northeast China. Before 2015, the IE in the northeast promoted the increase in FU. In 2016, the state proposed to strengthen soil environmental protection on agricultural land and promote the green development of agriculture in Northeast China. The decrease in fertilizer intensity in Northeast China made it the dominant force of FU reduction, accounting for 67.4% of the total reduction in China.

**3.2.4 Structure effect based on region.** It should be noted in Fig 6 and Fig 5D that the regional STE was steady during 2006–2017. However, the negative STE of each province strengthened with each piece, and this pattern caused the reduction trend to be visible. In Table 1, the number of provinces with a negative STE in the four periods was 16, 16, 19 and 21, respectively, which indicated that more provinces used a lower proportion of the national fertilizer application. This change was due to the increasing importance of FU in these provinces, and possible reduction measures have had some effect.

Generally, a positive STE was most prominent and increased most in western regions from 2006 to 2014. The agricultural production in the western part was rough, and FU could not be adequately controlled. However, the eastern region immensely promoted straw returning technology, formulated fertilizer and organic fertilizer. Therefore, the eastern region had a negative STE during this period, especially in Fujian (-53.6×104t), Guangdong (-33.9×104 t) and Hebei (-32.6×104 t), which played an active role in supporting fertilizer loss. In the central and northeastern regions, the STE was not significant, and the contribution to the change in fertilizer application rate was not prominent.

## 3.3 Driving factors based on fertilizer type affecting fertilizer reduction

**3.3.1 Total effect based on fertilizer type.** Fig 7A shows the changes in FU caused by the SE, IE and STE of the FU based on different fertilizer types. It is worth noting that the STE was virtually zero during the entire period of 2006–2017. Did the structure of FU not affect the FU? No, that does not make sense. This is the result of the interaction of different kinds of fertilizers (specific to the analysis in Fig 7D).

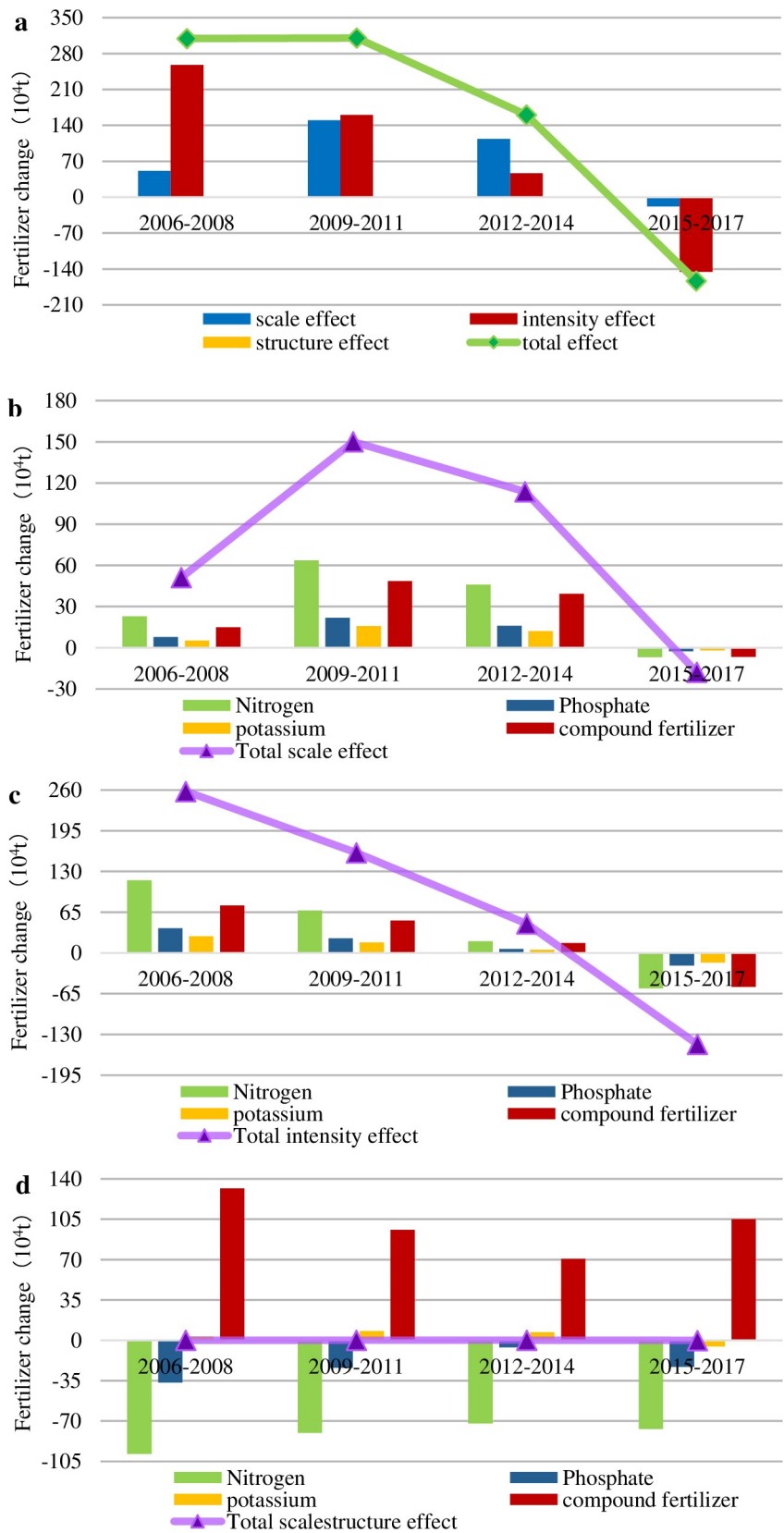

**Fig 7. The total effect, scale effect, intensity effect and structure effect based on fertilizer types perspective.** (a) Chinese FU variation decomposition based on fertilizer kinds; (b) Contribution of four fertilizers to scale effect; (c) Contribution of four fertilizers to intensity effect; (d) Contribution of four fertilizers to structure effect. As the total effect decomposition is different from the above two perspectives, so decomposed again.

In 2006–2008 and 2009–2011, China's FU maintained high growth. Nevertheless, the cause for the increase in FU during these two periods was distinguishing. The IE supported the increase in FU with an absolute advantage, while the SE served only as the auxiliary factor of fertilizer increment during 2006–2008. During the period between 2009 and 2011, the SE decreased, and the two effects were almost the same, promoting the growth of fertilizer application. However, from 2012 to 2014, the SE and IE declined, resulting in a downward trend in fertilizer increments. In 2015, the Ministry of Agriculture issued the National Agricultural and Rural Economic Development Plan, which propelled the protection and conservation of cultivated land and controlled the harm of fertilizer to the soil. Consequently, from 2015 to 2017, the SE and IE changed qualitatively, from a positive effect to a negative effect, contributing to the reduction in FU, but the contribution of the SE was far less than that of the IE.

**3.3.2 Effects based on fertilizer type.** There is a visible pattern seen in Fig 7B and Fig 7C, and the position of each fertilizer is invariable in both SE and IE. The SE and IE of nitrogen fertilizer were the largest, followed by compound fertilizer, which accounted for 70% of the SE and IE. The SE and IE of phosphate fertilizer and potash fertilizer were relatively inappreciable.

In addition, the STE of different fertilizers differed significantly, as shown in Fig 7D. Compound fertilizers and nitrogen fertilizers had the most substantial contributions to the STE. During 2006–2017, the compound fertilizers had a positive STE, while nitrogen fertilizers had a negative STE. In contrast, phosphorus and potassium fertilizers contributed less to the structural effects. The positive and negative STE of the four fertilizers cancelled each other out, which showed that the total STE was zero. This result was mainly due to the unreasonable structure of fertilizer application in China. Nitrogen application was the highest, accounting for approximately 40% of the total FU, while nitrogen and phosphate applications were less, accounting for only approximately 25% of the total FU (Fig 8). The excessive use of nitrogen fertilizer can easily lead to low fertilizer efficiency, hinder mineral nutritional activity, harm crop growth and cause other problems. Fortunately, the Chinese government has recognized these risks and has issued the Chemical Industry and the 12th Five-Year Plan for the Development of the Chemical Fertilizer Industry, encouraging the use of compound fertilizer and balancing the application of nitrogen, phosphorus and potassium fertilizers. Therefore, the proportion of nitrogen fertilizer application in China decreased from 45.91% to 37.92% in 2017, and the proportion of compound fertilizer application increased from 28.13% in 2006 to 37.89% in 2017. The fertilization structure has been continuously rationalized.

## 3.4 Study limitations

The limitation of this paper comes from a lack of data. (1) There is no first-hand data on FU per crop; thus, we must obtain it from the FU per mu × crop sown area. This algorithm regards the fertilizer application intensity as an average, which may be different from the actual value. (2) By referring to the official website of National Bureau of Statistics, the official website of the Ministry of Agriculture and Rural Affairs of The People's Republic of China, China Statistical Yearbook, China Rural Statistical Yearbook, National Compilation of Data on The Cost and Benefit of Agricultural Products and other Yearbooks related to agriculture, we did not find the data of " the fertilization area of each fertilizer ". However, nitrogen, phosphorus and

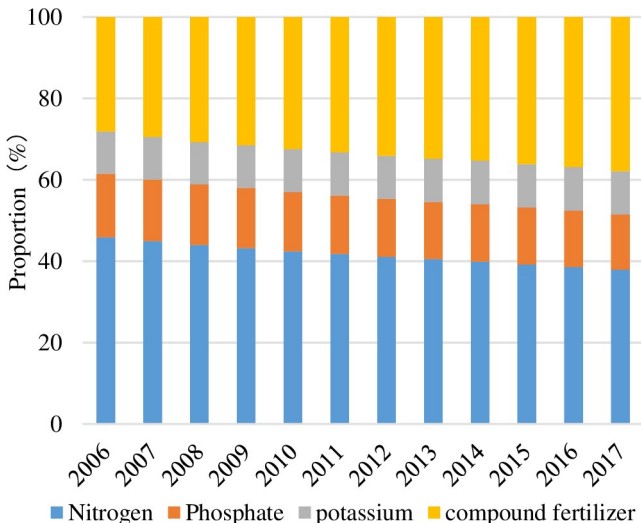

**Fig 8. The proportion of FU of four kinds of fertilizer during 2006–2017.**

potassium are almost essential elements in the growth process of all crops. Chinese agricultural operators often cross use a variety of fertilizers to ensure the growth of crops in the agricultural production process. Although the compound fertilizer contains three elements, other fertilizers will still be applied in the actual production process (refer to National Compilation of Data on The Cost and Benefit of Agricultural Products). In addition, according to the indicators on the official website of China Statistics Bureau, the application amount of four kinds of fertilizers used in this paper is narrow sense agricultural data (excluding forestry, animal husbandry and fishery, source: website of National Statistics Bureau). Therefore, we think that the fertilization area of nitrogen, phosphorus, potassium and compound fertilizer is about equal to the planting area of crops. (3) The statistical caliber of data from different sources is different. There is an individual error between the total sowing area of each province and the total sowing area of the whole country. Therefore, we need to discuss the effect of the total SE on the weight loss of fertilizers in both regional and fertilizer aspects.

## 4. Conclusions

Based on the panel data of FU in China from 2006 to 2017, this paper used the LMDI decomposition method to analyze the SE, IE and STE of the FU decline from the three perspectives of crops, regions and fertilizer types and probed the contribution of each effect, not only to provide the scientific basis for continuous reduction of fertilizer, but also to improve the management system of fertilizer reduction for policymakers and realize the reduction "Zero growth of FU" provides policy reference.

We found that the effect was declining in each perspective, and most of the effects changed from positive to negative, stimulating the continuous decline in FU. The IE was the leading driving factor affecting the decline in FU in China, and the contribution of the STE was also relatively significant, while the impact of the SE was the smallest. From the crop perspective, grain, vegetables and fruits contributed most to the decrease in FU, mainly in scale and intensity. However, the STE of tobacco was positive, which promoted the growth of FU. From a regional perspective, the situation of fertilizer weight loss in different regions was discrepant. The maximal contribution to fertilizer weight loss was the decrease in the SE and STE in the eastern area, while it was the fertilizer decrement in the central, western and northeastern

regions that mainly came from the decrease in the IE. From a fertilizer type perspective, nitrogen fertilizer and compound fertilizer were the two most commonly used fertilizers in China, and their SE and IE values decreased, playing a positive role in reducing the amount of fertilizer. In terms of the STE, the effect of compound fertilizer on fertilizer was opposed to the effect of nitrogen fertilizer. China's use of potash and phosphate fertilizer was less, contributing less to the reduction in fertilizer. According to the results of the analysis, China's FU is on a downward trend in the future. Therefore, we have reason to believe that FU may continue to decline.

How to ensure the continuous decline of FU? Through the factor decomposition analysis in this paper, we realize that the reduction of fertilizer comes from the common measures of fertilization area, fertilization intensity and fertilization structure. Therefore, this study believes that only the establishment of "reducing the intensity of fertilizer application, optimizing the planting structure and fertilizer type usage structure and stabilizing the planting area" of fertilizer reduction management system can guarantee the long-term, stable and sustained reduction of FU. Otherwise, there may be a rebound.

## Author Contributions

**Conceptualization:** Yuanmeng Ji, Huajun Liu.

**Data curation:** Yuanmeng Ji.

**Formal analysis:** Yuanmeng Ji, Huajun Liu, Yin Shi.

**Methodology:** Yuanmeng Ji, Yin Shi.

**Writing – original draft:** Yuanmeng Ji.

**Writing – review & editing:** Yuanmeng Ji, Huajun Liu.

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
