## [Decision Letter · Decision Letter 0]

10 Jun 2020

PONE-D-20-12238

Will China's fertilizer use continue to decline? Evidence from LMDI analysis based on crops, regions and fertilizer types.

PLOS ONE

Dear Dr. Liu,

Thank you for submitting your manuscript to PLOS ONE. After careful consideration, we feel that it has merit but does not fully meet PLOS ONE’s publication criteria as it currently stands. Therefore, we invite you to submit a revised version of the manuscript that addresses the points raised during the review process.

We look forward to receiving your revised manuscript.

Kind regards,

Bing Xue, Ph.D.

Academic Editor

PLOS ONE

Journal Requirements:

Reviewers' comments:

Reviewer's Responses to Questions

**Comments to the Author**

1. Is the manuscript technically sound, and do the data support the conclusions?

Reviewer #1: Yes

Reviewer #2: Yes

2. Has the statistical analysis been performed appropriately and rigorously? 

Reviewer #1: Yes

Reviewer #2: Yes

3. Have the authors made all data underlying the findings in their manuscript fully available?

Reviewer #1: Yes

Reviewer #2: Yes

4. Is the manuscript presented in an intelligible fashion and written in standard English?

Reviewer #1: No

Reviewer #2: Yes

5. Review Comments to the Author

Reviewer #1: China implemented the Action Plan for the Zero Increase of Fertilizer

Usein 2015, which led to adecrease infertilizer use. However, there is little quantitative research

on whether this decline will continue. With data obtained from 2006 to 2017, the paper used

the logarithmic mean Divisia index (LMDI) method to analyze the driving factors of fertilizer use

change in China from three aspects: crops, regions and fertilizer types. Furthermore, this

decomposition method provided us with the specific effects of fertilizer reduction, including the

scale effect, intensity effect and structural effect. The finding suggests that (1)the intensity effect

was the most critical factor affecting the decline in fertilizer use in China, followed by the scale

effect. Structure effects had the smallest impact.(2) The sowing scale and fertilization intensity. In reviewer opinion, the paper be revised as follows:\\\\

1. the abstract is too long and it should be shorted into 30 percent of it's now .\\\\

2. the contributions to application of this paper are not clear.?\\\\

3. the motivation of this paper should be further emphized.

Reviewer #2: Some modification suggestions：(1) With data obtained from 2006 to 2017, the paper used the logarithmic mean Divisia index (LMDI) method to analyze the driving factors of fertilizer use change in China from three aspects: crops, regions and fertilizer types, some useful conclusions are obtained, for example，the government should establish a long-term mechanism to control the scale, reduce the intensity and adjust the structure. Nevertheless, what is a long-term mechanism to control the scale, reduce the intensity and adjust the structure? The author's conclusion is very vague. (2) In fact, we would like to know if the reduction of fertilizer use in China will lead to the decrease of crop yield? The author should increase the content of this research. (3) The authors regard the fertilization area of each fertilizer as the same, but in fact the fertilization area of each fertilizer is different, relevant data from the statistical yearbook could be obtained. The authors should carry out this work. (4) There are a few grammatical errors in the manuscript.

6. PLOS authors have the option to publish the peer review history of their article (what does this mean?). If published, this will include your full peer review and any attached files.

Reviewer #1: No

Reviewer #2: No

---

## [Author Response · Author response to Decision Letter 0]

1 Jul 2020

Response to reviewers

Dear Editors of PLOS ONE,

Thank you for your letter and editor's attention to our manuscript, entitled “Will China's fertilizer use continue to decline? Evidence from LMDI analysis based on crops, regions and fertilizer types”(PONE-D-20-12238). The comments are valuable and have been very helpful for revising and improving this study. In accordance with the requirements of PLOS ONE and the opinions of the reviewers, I revised the manuscript again. In addition, after discussion by all authors, I have adjusted the order of authors. On behalf of my colleagues and myself, I am resubmitting the manuscript to your journal. The main corrections to the paper and the responses to the comments of the academic editor and reviewers are appended below. The comments are presented in blue, and our responses are in black. The detailed changes can be found in the ‘Manuscript’ and ‘Revised Manuscript with Tracked Changes’.

Yours sincerely,

Huajun Liu, PhD

Professor, Ph. D. supervisor, Shandong University of Finance and Economics.

 

Response to Reviewer 1’s Comments

PONE-D-20-12238

Will China's fertilizer use continue to decline? Evidence from LMDI analysis based on crops, regions and fertilizer types

25- June -2020

Dear reviewer 1:

Thank you for your constructive comments on our manuscript, which have helped us to further revise and improve our paper. Based on your recommendations, we have modified the manuscript carefully. Reviewer 1’s suggestions are shown in blue, and our responses are shown in black. In addition, the corresponding modifications are shown by tracked changes in the manuscript. The main corrections and the responses to the reviewer’s comments are shown as follows:

 The abstract is too long and it should be shorted into 30 percent of it's now.

We are grateful for your constructive suggestion. We have simplified the Abstract according to your requirements and kept the key conclusions. At present, we have deleted 245 words to 176 words. The specific changes are as follows: 

China implemented the Action Plan for the Zero Increase of Fertilizer Use in 2015, which led to a decrease in fertilizer use. However, Will fertilizer use continue to reduce? With data obtained from 2006 to 2017, the paper used the logarithmic mean Divisia index (LMDI) method to analyze the scale effect, intensity effect and structural effect of fertilizer use change in China from three aspects: crops, regions and fertilizer types. Our finding suggests that (1) The intensity effect was the most critical factor affecting the decline in fertilizer use in China. (2) The sowing scale and fertilization intensity of grain, vegetables and fruits had the most significant driving effect on fertilizer reduction. (3) The three effects of each region were different in space, and the eastern region contributed most to the fertilizer decrement. (4) Nitrogen fertilizer and compound fertilizer had the most considerable influence on fertilizer reduction, especially in sowing scale and fertilization intensity since 2009. The government should establish a fertilizer reduction management system, which includes scale control, intensity reduction, structural adjustment and other measures.

 The contributions to application of this paper are not clear.

Thank you for this helpful suggestion. In the fifth paragraph of the Introduction and the first paragraph of the Conclusion, we have made the contributions to application of this paper more clearly. The specific changes are shown in the following red font:

 In the fifth paragraph of the Introduction 

Based on the existing research, the main contributions of this paper are two aspects. First, this paper expands the perspective of fertilizer research, and discusses the sources of fertilizer reduction from the perspectives of crop, region and fertilizer type. Second, this paper is the first attempt to answer the question of sustainability of fertilizer reduction in China. Therefore, the paper used LMDI method to decompose the driving factors of the change of FU in 2006-2017 from three aspects of crop, region and fertilizer type, and deeply explores the sources of fertilizer reduction in China from different perspectives. The purpose of the study is not only to provide scientific reference for better reducing usage, increasing efficiency of fertilizer and controlling the excessive use of fertilizer, but also to provide targeted policy suggestions for exploring modern fertilizer management and achieving the goal of "zero growth of FU" in China.

 In first paragraph of the Conclusion

 Based on the panel data of FU in China from 2006 to 2017, this paper used the LMDI decomposition method to analyze the SE, IE and STE of the FU decline from the three perspectives of crops, regions and fertilizer types and probed the contribution of each effect, not only to provide scientific basis for continuous reduction of fertilizer, but also to improve the management system of fertilizer reduction for policy makers and realize the reduction "Zero growth of FU" provides policy reference.

 The motivation of this paper should be further emphized.

Thank you for this valuable suggestion. We have emphasized the motivation of this paper in the Abstract, the second and fifth paragraphs of the Introduction. The specific changes are shown in the following red font:

 In Abstract

China implemented the Action Plan for the Zero Increase of Fertilizer Use in 2015, which led to a decrease in fertilizer use. However, Will fertilizer use continue to reduce? With data obtained from 2006 to 2017, the paper used the logarithmic mean Divisia index (LMDI) method to analyze the scale effect, intensity effect and structural effect of fertilizer use change in China from three aspects: crops, regions and fertilizer types. Our finding suggests that (1) The intensity effect was the most critical factor affecting the decline in fertilizer use in China. (2) The sowing scale and fertilization intensity of grain, vegetables and fruits had the most significant driving effect on fertilizer reduction. (3) The three effects of each region were different in space, and the eastern region contributed most to the fertilizer decrement. (4) Nitrogen fertilizer and compound fertilizer had the most considerable influence on fertilizer reduction, especially in sowing scale and fertilization intensity since 2009. The government should establish a fertilizer reduction management system, which includes scale control, intensity reduction, structural adjustment and other measures.

 In the second paragraph of the Introduction

In recent years, the Chinese government has recognized the seriousness of the overuse of fertilizer. It has put forward the decision of reducing the amount of fertilizers and increasing the efficiency on the premise of stable food production growth and adequate protection of food security. In 2015, the Chinese government promulgated the Action Plan for the Zero Increase of FU, which proposed a goal of "zero growth of FU, the establishment of a scientific fertilizer management technology system, and the improvement of the scientific FU level". Then, in 2016 and 2017, Central Document No.1 noted that the "zero growth" action of fertilizer should be carried out. China's zero-growth action in FU has achieved initial results. In 2016, China's FU approached zero growth for the first time. FU in China declined from 60.226 million tons in 2015 to 58.59 million tons in 2017, with an annual rate of decline of 1.8% (Fig. 1). Then, will the decline in China's FU be sustainable? Further research on this question not only helps us better explore the driving effects of China's fertilizer reduction and influence of each effects but also provides more comprehensive reference for policymakers to continuously control the fertilizer decrement and develop a sound fertilizer reduction management system.

 In fifth paragraph of the Introduction

Based on the existing research, the main contributions of this paper are two aspects. First, this paper expands the perspective of fertilizer research, and discusses the sources of fertilizer reduction from the perspectives of crop, region and fertilizer type. Second, this paper is the first attempt to answer the question of sustainability of fertilizer reduction in China. Therefore, the paper used LMDI method to decompose the driving factors of the change of FU in 2006-2017 from three aspects of crop, region and fertilizer type, and deeply explores the sources of fertilizer reduction in China from different perspectives. The purpose of the study is not only to provide scientific reference for better reducing usage, increasing efficiency of fertilizer and controlling the excessive use of fertilizer, but also to provide targeted policy suggestions for exploring modern fertilizer management and achieving the goal of "zero growth of FU" in China.

 

Response to Reviewer 2’s Comments

PONE-D-20-12238

Will China's fertilizer use continue to decline? Evidence from LMDI analysis based on crops, regions and fertilizer types

25- June -2020

Dear reviewer 2:

We are grateful for your constructive suggestions about our manuscript, which have helped us to further revise and improve the paper. Based on your recommendations, we have modified the manuscript carefully. Reviewer 2’s suggestions are shown in blue, and our responses are shown in black. In addition, the corresponding modifications are expressed by tracked changes in the manuscript. The main corrections and the responses to the reviewer’s comments are shown as follows:

(1) With data obtained from 2006 to 2017, the paper used the logarithmic mean Divisia index (LMDI) method to analyze the driving factors of fertilizer use change in China from three aspects: crops, regions and fertilizer types, some useful conclusions are obtained, for example，the government should establish a long-term mechanism to control the scale, reduce the intensity and adjust the structure. Nevertheless, what is a long-term mechanism to control the scale, reduce the intensity and adjust the structure? The author's conclusion is very vague.

Response of the authors:

Thanks for your suggestion. The "long-term mechanism" mentioned by the author in the Abstract is misexpression. A better way of expression is not a long-term mechanism of fertilizer reduction, but a management system of fertilizer reduction. We apologize and once again thank the reviewers for this enlightening suggestion. We may continue to write articles in this field in the future, and we will seriously explore this issue. Then, in Abstract and Conclusions, we have made a supplementary explanation. The specific changes are shown in the following red font:

(1) In Abstract

China implemented the Action Plan for the Zero Increase of Fertilizer Use in 2015, which led to a decrease in fertilizer use. However, Will fertilizer use continue to reduce? With data obtained from 2006 to 2017, the paper used the logarithmic mean Divisia index (LMDI) method to analyze the scale effect, intensity effect and structural effect of fertilizer use change in China from three aspects: crops, regions and fertilizer types. Our finding suggests that (1) the intensity effect was the most critical factor affecting the decline in fertilizer use in China. (2) The sowing scale and fertilization intensity of grain, vegetables and fruits had the most significant driving effect on fertilizer reduction. (3) The three effects of each region were different in space, and the eastern region contributed most to the fertilizer decrement. (4) Nitrogen fertilizer and compound fertilizer had the most considerable influence on fertilizer reduction, especially in sowing scale and fertilization intensity since 2009. The government should establish a fertilizer reduction management system, which includes scale control, intensity reduction, structural adjustment and other measures.

(2) In Conclusions

How to ensure the continuous decline of FU? Through the factor decomposition analysis in this paper, we realize that the reduction of fertilizer comes from the common measures of fertilization area, fertilization intensity and fertilization structure. Therefore, this study believes that only the establishment of "reducing the intensity of fertilizer application, optimizing the planting structure and fertilizer type usage structure and stabilizing the planting area" of fertilizer reduction management system can guarantee the long-term, stable and sustained reduction of FU. Otherwise, there may be a rebound.

(2) In fact, we would like to know if the reduction of fertilizer use in China will lead to the decrease of crop yield? The author should increase the content of this research.

Response of the authors:

Thank you for your constructive and helpful suggestions. We have increased the changing trend of the fertilization intensity of eight crops and unit yield from 2006 to 2017 to illustrate this problem in 3.1.5. The details are as follows.

3.1.5 The yield of crops during fertilizer reduction

In recent years, the three effects have driven the decline of FU in varying degrees. Especially after 2015, China's FU has successfully decreased year after year. On the contrary, China's food production has not been reduced, but continued to rise (Fig. 1). So, from the perspective of crops, will the decrease of FU lead to the decrease of crop yield?

Fig. 2 shows that the IE is the most important factor leading to the change of crop FU, and also the main source of the decrease of FU, while vegetables, grain and fruits are the main crops causing the decrease of IE. It can be seen from the Fig. 3 that the fertilization intensity of vegetables, grain, sugar, beans and cotton has been reduced, especially grain and vegetables. However, the unit yield of crops has not declined as a result, showing a continuous growth phenomenon. The main reasons for this are as follows. First, there is a general phenomenon of excessive fertilization in China's agriculture. Therefore, properly reducing the intensity of fertilization will not lead to the loss of nutrients in crops, which will not threaten crop yield. Second, according to different fertilization methods, China has developed some mature technical models, especially the promotion of high-yield and high-efficiency cultivation technology model, which can not only reduce the fertilizer intensity, but also increase the per unit yield to a certain extent. Third, the popularization of soil testing formula fertilization technology also plays a role in saving fertilizer and increasing production. In addition, although the fertilization intensity of fruits fluctuated greatly, it did not affect the growth of per unit yield. In the next step, we should continue to control the fertilization intensity of fruits. The fertilization intensity of oils is very similar to the change trend of per unit yield, and the application intensity of fertilizer is likely to have a high impact on per unit yield. Therefore, replacing conventional materials with new fertilizers may achieve the reduction of fertilizer without affecting per unit yield. It is worth noting that after 2009, the fertilization intensity and yield of tobacco changed in the opposite direction, indicating that the decline of fertilization intensity of tobacco will not directly lead to the decline of yield. In this view, China's FU reduction action is implemented under the condition of ensuring food security or crop production security. Fertilizer reduction will not lead to crop production reduction.

Fig.1. Food production and FU in China during 2003-2017.

Fig. 2. The total effect, scale effect, intensity effect and structure effect based on crop perspective: (a) Chinese FU variation decomposition based on crops;(b) Contributions of key crops to scale effect; (c) Contributions of key crops to intensity effect;(d) Contributions of key crops to structure effect.

Fig. 3. Per unit yield and fertilization intensity of eight crops

(3) The authors regard the fertilization area of each fertilizer as the same, but in fact the fertilization area of each fertilizer is different, relevant data from the statistical yearbook could be obtained. The authors should carry out this work.

Response of the authors:

Thank you for this valuable suggestion. According to your request, we first carefully looked up the official website of China Statistics Bureau, the official website of the Ministry of Agriculture and Rural Affairs of The People's Republic of China, China Statistical Yearbook, China Rural Statistical Yearbook, National Compilation of Data on The Cost and Benefit of Agricultural Products and other yearbooks related to agriculture, but we did not find the data of " the fertilization area of each fertilizer ". The reason why the author thinks that the fertilization area of each fertilizer is equal is that nitrogen, phosphorus and potassium are almost the essential elements in the growth process of all crops. In the agricultural production process, Chinese agricultural operators often cross use a variety of fertilizers to ensure the growth of crops. Although the compound fertilizer contains three elements, other fertilizers will still be applied in the actual production process (refer to National Compilation of Data on The Cost and Benefit of Agricultural Products). In addition, according to the indicators on the official website of China Statistics Bureau, the application amount of four kinds of fertilizers used in this paper is narrow sense agricultural data (excluding forestry, animal husbandry and fishery, source: website of National Statistics Bureau). We think this data is also reasonable, that is, all the fertilizers are used on the seeded land. Therefore, we think that the fertilization area of nitrogen, phosphorus, potassium and compound fertilizer is about equal to the planting area of crops, but this method also has some defects. Specific instructions are also explained in 2.2 and 3.4 of the manuscript.

Then, based on LMDI decomposition formula, we try to estimate the fertilization area of each fertilizer.

where, F is the FU of 4 fertilizers, and k represents the fertilizer types (nitrogen fertilizer, phosphate fertilizer, potash fertilizer and compound fertilizer). f_k is the FU of fertilizer k，S_k is the fertilized area of fertilizer k and S is the total sown area of 4 fertilizers. f_k⁄S_k , S_k⁄S and S are represented intensity factor, structure factor and scale factor. The specific steps are as follows. First, from the official website of China Statistics Bureau, we found the data of "the use amount of national nitrogen, phosphorus, potassium and compound fertilizer". Secondly, from the National Compilation of Data on The Cost and Benefit of Agricultural Products, we found the intensity of applying nitrogen, phosphorus, potassium and compound fertilizers to different crops in China. The intensity of applying nitrogen, phosphorus, potassium and compound fertilizer to crops in China is estimated by calculating the average intensity of applying nitrogen, phosphorus, potassium and compound fertilizer to grains, beans, oil plants, sugar, cotton, tobacco, fruits and vegetables. Third, calculate the fertilizer application area. For example, Application area of nitrogen = application amount of nitrogen (in step1) / application intensity of nitrogen (in step2). But this method has some defects. First, it is questionable to use this method to calculate the fertilizer application area, because up to now we have not found any scholars using the same method. Second, although we can estimate the application area of each fertilizer, we cannot estimate the total application area required by the formula. At this time, we can not use the national planting area to replace the total fertilization area of four kinds of fertilizers. For example, the annual average fertilized area of the estimated phosphorus fertilizer is nearly 10000 thousand hectares higher than the actual national sown area. Therefore, considering many aspects, we give up the estimation method and keep the previous choice.

 (4) There are a few grammatical errors in the manuscript.

Thank you for this helpful suggestion. We have revised the English carefully, and asked a competent editor to review the English. The Editing Certificate has been submitted as a separate document. If there are still problems, we will continue to modify and improve it.

---

## [Decision Letter · Decision Letter 1]

23 Jul 2020

Will China's fertilizer use continue to decline? Evidence from LMDI analysis based on crops, regions and fertilizer types.

PONE-D-20-12238R1

Dear Dr. Liu,

We’re pleased to inform you that your manuscript has been judged scientifically suitable for publication and will be formally accepted for publication once it meets all outstanding technical requirements.

Kind regards,

Bing Xue, Ph.D.

Academic Editor

PLOS ONE

Additional Editor Comments (optional):

Reviewers' comments:

Reviewer's Responses to Questions

**Comments to the Author**

1. If the authors have adequately addressed your comments raised in a previous round of review and you feel that this manuscript is now acceptable for publication, you may indicate that here to bypass the “Comments to the Author” section, enter your conflict of interest statement in the “Confidential to Editor” section, and submit your "Accept" recommendation.

Reviewer #1: All comments have been addressed

Reviewer #2: All comments have been addressed

2. Is the manuscript technically sound, and do the data support the conclusions?

Reviewer #1: Yes

Reviewer #2: Yes

3. Has the statistical analysis been performed appropriately and rigorously? 

Reviewer #1: Yes

Reviewer #2: Yes

4. Have the authors made all data underlying the findings in their manuscript fully available?

Reviewer #1: Yes

Reviewer #2: Yes

5. Is the manuscript presented in an intelligible fashion and written in standard English?

Reviewer #1: Yes

Reviewer #2: Yes

6. Review Comments to the Author

Reviewer #1: the paper has been revised and the revision basicly satisfies the requirements of the reviewer, hence i recommend it should be accepted to publish in the journal. Thank you

Reviewer #2: The author answered my questions and revised the manuscript very well. I think it meets the publishing requirements.

7. PLOS authors have the option to publish the peer review history of their article (what does this mean?). If published, this will include your full peer review and any attached files.

Reviewer #1: No

Reviewer #2: No

---

## [Editor Report · Acceptance letter]

3 Aug 2020

PONE-D-20-12238R1 

Will China's fertilizer use continue to decline? Evidence from LMDI analysis based on crops, regions and fertilizer types. 

Dear Dr. Liu:

I'm pleased to inform you that your manuscript has been deemed suitable for publication in PLOS ONE. Congratulations! Your manuscript is now with our production department. 

Kind regards, 

on behalf of

Professor Bing Xue 

Academic Editor

PLOS ONE